# A Review of Non-Invasive Skin Imaging in Merkel Cell Carcinoma: Diagnostic Utility and Clinical Implications

**DOI:** 10.3390/cancers16213586

**Published:** 2024-10-24

**Authors:** Iulia Maria Badiu, Katarzyna Korecka, Anca Olguta Orzan, Marco Spadafora, Caterina Longo, Ana-Maria Forsea, Aimilios Lallas

**Affiliations:** 1Department of Oncologic Dermatology, Elias University Hospital, Carol Davila University of Medicine and Pharmacy, 050474 Bucharest, Romania; olguta.orzan@umfcd.ro (A.O.O.); ana.forsea@umfcd.ro (A.-M.F.); 2Department of Dermatology and Venereology, Poznań University of Medical Sciences, 61-701 Poznań, Poland; kkorecka@ump.edu.pl; 3Centro Oncologico ad Alta Tecnologia Diagnostica, Azienda Unità Sanitaria Locale—IRCCS di Reggio Emilia, 42122 Reggio Emilia, Italy; marco.spadafora@unimore.it (M.S.); caterina.longo@unimore.it (C.L.); 4Department of Dermatology, University of Modena and Reggio Emilia, 41121 Modena, Italy; 5First Dermatology Department, School of Medicine, Faculty of Health Sciences, Aristotle University, GR-54124 Thessaloniki, Greece; alallas@auth.gr

**Keywords:** Merkel cell carcinoma, non-invasive, skin imaging, nonmelanoma skin cancer, dermoscopy, reflectance confocal microscopy, high-frequency ultrasound

## Abstract

Merkel cell carcinoma (MCC) is a rare and aggressive skin cancer that grows quickly and has a life-threatening potential. Because of its severity, it is important to provide a fast diagnostic. For a tumor like this, diagnosis usually requires a tissue sample examined under a microscope with special staining techniques. However, newer non-invasive imaging methods could help identify MCC without a biopsy. These include advanced techniques which provide detailed images of the skin. This review highlights recent research on these imaging methods, emphasizing their potential to improve MCC diagnosis and treatment planning. This can also help patients get diagnosed more quickly and comfortably, without the need for more invasive procedures.

## 1. Introduction

Merkel cell carcinoma (MCC) is a rare but aggressive malignancy that affects the skin, has a rapid and unpredictable growth, and usually develops on sun-exposed areas like the face, neck, or arms. A major pathogenetic factor of MCC is the so-called Merkel cell polyoma virus (MCPyV), although the precise role of the virus in the carcinogenetic pathway has not been fully elucidated [1,2]. Approximately 80% of MCCs are associated with MCPyV. Another major pathogenetic factor is long-term sun exposure, which is involved in both viral-mediated and non-viral-mediated carcinogenesis, by contributing to immunosuppression or DNA damage, respectively [1,2,3]. Both MCC types exhibit changes in the structure and function of the retinoblastoma protein and p53 gene [1].

Individuals with a weakened immune system are at a higher risk of developing MCC and so are older individuals, especially those over the age of 50. Additional factors such as a fair phototype or a history of chronic diseases might also increase the risk of developing MCC [2].

The tumor usually appears as a firm flesh-colored or bluish-red nodule, and it can easily mimic other benign and malignant skin tumors (Figure 1 and Figure 2).

MCC is a very aggressive tumor, with lymph node involvement found in 26% of cases at diagnosis and distant metastases in 8% of cases [1].

Early-stage MCC is typically managed with surgery and adjuvant radiation, while advanced or metastatic MCC requires systemic therapies, mainly immunotherapy with anti-programmed cell death-1 antibodies. According to current guidelines, sentinel lymph node biopsy is advised for all MCC patients with tumors > 1 cm in diameter [3,4].

Non-invasive modalities such as dermoscopy, reflectance confocal microscopy (RCM), optical coherence tomography (OCT), high-frequency ultrasound (HFUS), in vivo reflectance and fluorescence spectroscopy, multispectral imaging and thermography could be valuable tools in the diagnosis and treatment evaluation of MCC. These imaging techniques can help detect MCC at an earlier stage, differentiate it from benign tumors, assess tumor extension, and monitor tumor recurrence or progression, enhancing overall management (Table 1). Due to the rarity of this tumor; however, no algorithms of non-invasive imaging diagnosis could be established. In this narrative review, we aim to offer an overview of the currently available methods for non-invasive exploration of MCCs, highlighting the advantages and disadvantages of each, in order to provide clinicians with a practical orientation for the optimal case-adapted approach.

This review is based on a search of the literature performed in PubMed for scientific evidence, including all studies regarding Merkel cell carcinoma and the non-invasive methods described below that were used for diagnostic purposes until 2024. The keywords selected were “Merkel cell Carcinoma” and any of the following: “diagnosis, imaging, high frequency ultrasound, dermoscopy, confocal microscopy, ultrasound, optical coherence tomography, multispectral, non-invasive, imaging”. The search was limited to works in English, and to studies on human subjects. Titles and abstracts were checked for eligibility. The references of selected relevant articles were retrieved and included in the analysis if relevant.

## 2. Dermoscopy of MCC

Dermoscopy (Table 2) is a non-invasive diagnostic method that allows for the examination of morphological details that cannot be seen with the naked eye, therefore enhancing the visualization of skin features in a non-invasive, accessible way. The correlation between dermoscopy aspects and histopathological structures of skin and skin lesions has been extensively described, and [3] this simple-to-use, accessible, non-invasive diagnostic technique has become widely used in dermatological practice for diagnosis and management of skin tumors, as well as other many other skin conditions such as psoriasis, eczema, and skin infections [5]. The dermoscope is a portable diagnostic device featuring a magnifying lens (typically ranging from 3× to 20×) and a polarized or nonpolarized light source. It enables en face visualization of epidermal and dermal structures that are otherwise imperceptible to the naked eye [4,5]. The technique has been greatly expanded by the addition of digital capture of images, and consecutively by the development of software for image capture, storage, mapping, analysis, and, lately, AI-based automatic diagnosis [6,7,8].

Dermatoscopic examination of MCCs (Figure 3 and Figure 4) typically shows a prominent red color, corresponding to either a dense vascularity or widespread erythema [1]. The most frequently observed dermoscopic features of MCCs are milky red (pink) areas along with some vascular anomalies, as described by Sadeghinia et al. [9]. Pink color was reported in most studies on MCC, either observed as a pink structureless background or as smaller round areas (milky red zones, globules, or clods) [3,9,10,11]. Although pink color can also be found in amelanotic melanoma, a lack of pigmentation or a blue-gray veil point towards a probable MCC [10,11,12,13]. White areas are also commonly reported in MCCs [10]. In terms of vascular pattern, linear irregular vessels, dotted vessels, and a polymorphous vascular pattern have been described [3,9,10,14,15]. However, these structures are present in other tumors as well, such as amelanotic melanoma or poorly differentiated squamous cell carcinoma. Arborizing vessels or branching vessels can also be seen in MCCs, rendering discrimination from BCC plurally challenging [12]. Less frequent dermoscopic features of MCCs are white scales and hyperkeratosis [13].

A study by Matthew J Lin et al. [11] suggested that dermoscopy is a helpful tool in the differential diagnosis of pink nodules, distinguishing highly aggressive malignancies such as amelanotic melanoma and MCC from benign non-pigmented skin tumors. The study included 150 patients with nodular melanocytic lesions and concluded that dermoscopy has a higher specificity (of 89%) than naked-eye examination (of 67%) in the diagnosis of non-pigmented nodules. The most characteristic features of MCCs were polymorphous vessels, white lines, and white structureless areas. Some MCCs displayed ulceration and keratin scale, which were associated with aggressive tumor behavior [11].

Overall, the dermatoscopic pattern of MCC is rather unspecific as it shares characteristics with other non-pigmented skin tumors, and further research is required to elucidate whether dermoscopy can improve clinical recognition of MCCs. Nevertheless, the presence of diverse vessel patterns and/or milky red color raises the suspicion of malignancy because these features are extremely uncommon in benign tumors.

## 3. High-Frequency Ultrasound (HFUS)

HFUS, also known as high-resolution ultrasound, offers a visualization of the skin in a non-invasive and accessible way, producing real-time images that help to analyze all layers from the epidermis to the hypodermis [16]. The depth of visualization achieved by HFUS is approximately 1–3 mm, which is sufficient to assess the entire thickness of the skin and superficial lesions [17,18].

High-frequency ultrasound, also known as high-resolution ultrasound, allows visualization of the skin in a non-invasive and accessible way, producing real-time images that help analyze all layers, from the epidermis to the hypodermis [16]. The visualization of the skin through US requires higher frequencies than regular use of this technique for organs or soft-tissue exploration, in the range of 20 MHz to 50 MHz. For a more precise assessment of the epidermic layer, frequencies of 75–100 MHz will be needed [19].

Since 1980, when it was first used for differentiating the skin depth of normal patients from that of a group with cutaneous disorders [20], this technique has gained popularity in the field of dermatology for aiding in diagnosis and decision making in the further treatment of skin tumors, as well as inflammatory diseases such as psoriasis, atopic dermatitis, and connective tissue diseases [21,22]. In the last few years, HFUS has also become an important asset in cosmetic dermatology, providing guidance for the injection of dermal fillers in dangerous anatomical zones [23,24].

The predominant use of HFUS in dermatology is in the evaluation of non-melanoma and melanoma skin cancers, among which basal cell carcinoma and melanoma have been best described [20].

The most recently published study by Tiberiu Tamas et al. [23] included 31 patients with 32 skin lesions of the head and neck highly suggestive of non-melanoma skin cancer (NMSC). The study aimed to compare tumor parameters evaluated through ultrasonography before and after surgical resection. It concluded that this non-invasive method helps avoid unnecessary traumatic skin biopsies for NMSC. The study utilized HFUS with transducers of 13 MHz, 20 MHz, and 40 MHz, recording tumor dimensions, presence of necrosis, regional lymph node status, and vascularization. The measurements revealed a mean tumor thickness of 4.5 mm and a width of 8.3 mm. HFUS accurately identified tumor margins and depth, showing a mean difference of only 0.2 mm when compared to histopathological results.

Most studies that measure invasion depth using HFUS for NMSC have focused on BCCs. The literature currently lacks sufficient data regarding the features of MCCs in HFUS imaging.

A review by Plocka et al. described the appearance of MCC in HFUS as an ill-defined dermal lesion which has a tendency to infiltrate the subcutaneous layers [25]. MCC is depicted as a hypoechoic mass with some hyperechoic areas, showing posterior acoustic enlargement and a dense epidermis [17]. Doppler analysis reveals a highly vascular tumor. However, recurrent lesions appear to have a more discrete flow.

HFUS (Table 3) has the advantage of being a cost-effective, non-invasive imaging method that, unlike dermoscopy, can evaluate the thickness of a skin tumor prior to biopsy and guide the excision plan [25].

However, HFUS also has limitations. The hypoechoic appearance of most tumors on HFUS can make it challenging to distinguish between different types of skin cancers, as many malignant and benign lesions appear similar. Additionally, while HFUS can provide detailed images of the tumor, it may not always clearly delineate the tumor margins, especially in cases where the lesion infiltrates deeply into subcutaneous layers or when there is significant surrounding inflammation or fibrosis [25]. Moreover, the technique’s reliance on operator expertise and the quality of the ultrasound equipment can lead to variability in diagnostic accuracy. Therefore, combining HFUS with other imaging modalities, such as dermoscopy or RCM, can enhance diagnostic precision and provide a more comprehensive evaluation of MCC.

In addition, HFUS can identify MCC metastases and assess the state of lymph nodes, which is of high value in tumor staging [17].

## 4. Reflectance Confocal Microscopy (RCM)

Among the new non-invasive imaging methods that have emerged in the last few years in dermatology, RCM is probably the most well established for its capacity to enable real-time visualization of cellular and subcellular cutaneous structures. (Table 4). RCM has high diagnostic accuracy for skin tumors, with several studies illustrating a correspondence between RCM features and dermoscopic and histologic patterns [26,27,28].

Among the new non-invasive imaging methods that have emerged in the past few years in the realm of dermatology, reflectance confocal microscopy (RCM) has become prevalent for its capacity of enabling real-time visualization of cellular and subcellular cutaneous structures. RCM has a high diagnostic accuracy for skin tumors, with a great number of studies illustrating a correspondence of the confocal features with dermoscopic and histologic patterns [29,30,31].

RCM imaging offers resolution ranging from 0.5 to 1 μm, with an axial resolution falling within the range of 3 to 5 μm [18]. RCM operates by directing a low-power laser beam onto the tissue under examination. This focused laser beam penetrates the skin and, as it interacts with various cellular structures, including cell membranes, nuclei and other organelles, a portion of the light is reflected back to a detector. The detected light is then meticulously processed and reconstructed into real-time, high-resolution images by reflecting the light into electrical signals. This allows for the observation of cell morphology, architecture, and vasculature of the cutaneous tissue [16]. Skin imaging can be performed either directly on the patient or on excised skin. This technology provides detailed information, offering essentially a virtual biopsy, minimizing patient discomfort and risk, and guiding diagnosis and a further therapeutic plan [18].

There are very few studies of RCM for MCC. As most tumors are nodular, the method has limited value due to the restricted depth of penetration. Longo et al. [32] reported that the RCM patterns in a case of MCC correlated well with histologic features (Figure 5 and Figure 6). Clinically, the patient presented with a red nodule with unspecific dermoscopic features, including a disorganized vascular pattern and milky red structures. RCM revealed grouped hyporeflective cells interfering with a few bright cells, all surrounded by a fibrotic stroma. Visible blood vessels were seen next to the tumor’s proliferation. The differential diagnosis was BCC and melanoma, but the lack of specific features for these tumors raised the suspicion of MCC as well [33].

Navarrete-Dechent et al. [26] described a case of intraepidermal MCC using both dermoscopy and RCM. MCC affecting only the epidermic layer of the skin is very unusual, and most of these superficial MCCs appear in conjunction with other malignancies [27]. The clinical aspect was that of a scaly plaque which was first assessed as an extramammary Paget disease but the biopsy was not conclusive. Dermoscopy revealed scaly areas, dotted vessels, reddish zones, and shiny lines, while RCM illustrated pagetoid epidermic cells and glandular nests at the barrier between the dermis and epidermis. Immunohistochemistry results were compatible with the diagnosis of an intraepidermal MCC, showing the presence of CK20, INMS1, and chromogranin. The study concluded that intraepidermal MCC and classical MCC display different features in RCM because of their different origins (epidermis vs. dermis) [26].

In a study conducted by Cinotti et al. [28] in four patients, RCM features of MCCs and secondary tumors were similar to those described by Longo et al. [32]. The four patients presented with four main lesions and six metastases of MCCs, which were confirmed by a biopsy. Dermoscopy showed an irregular vascular pattern on a pink background in the primary tumors, while the secondary tumors displayed a rather purple background accompanied by white structures. RCM evaluation demonstrated dermal clusters of compact-size hyporeflective cells, similar in aspect to lymphocytes, delimited by bands of connective fibrous tissue. The epidermic layer had a small density and was disrupted by the tumors underneath. Unlike dermoscopy, RCM proved the same characteristics for both the primary and secondary tumors. According to the study, the element that could point towards the diagnosis of MCC over amelanotic melanoma is a more monomorphous appearance and a better-defined fibrotic stroma within the conglomeration of cells. A new feature that the study detected was a dissociation of cell aggregates in certain regions of the tumors, an observation that was strongly correlated with the histology results.

## 5. Optical Coherence Tomography (OCT)

OCT is an advanced, non-invasive imaging modality that utilizes principles of low-coherence interferometry to provide high-resolution, cross-sectional visualization of tissues (Table 5). OCT facilitates the envisioning of skin layers with exceptional detail, enabling the examination of epidermal and dermal structures at a micron-scale resolution, reaching a skin depth of 2 mm [34]. While OCT was initially employed in the realm of ophthalmology, where it remains utilized, its effectiveness has been proven in various disciplines and its application in dermatology is widely accepted currently [35].

The fundamental concept behind OCT imaging is similar to that of ultrasound imaging. Instead of utilizing acoustic waves, OCT measures echo delays and the intensity of back-reflected infrared light. While OCT provides more accurate visualization of skin lesions than HFUS, its depth of imaging is lower than that of RCM. OCT allows evaluation of the skin to a depth of 0.4 to 2.0 mm with an optical resolution ranging from 3 to 15 μm [17].

OCT is useful in the diagnosis of MNSC, as it reveals epidermal structure atypia, but also DEJ changes and dermal invasion, due to its deeper penetrance [34]. OCT also has the capacity to represent the epidermal malignant cellular clusters [36].

A study by Donelli et al. pointed out the significant role of line-field confocal coherence tomography (LC-OCT) regarding the diagnosis of NMSC, demonstrating a superior performance to that of dermoscopic examination, especially in the case of BCCs [37]. Cinotti et al. compared the efficacy of reflectance confocal microscopy (RCM) and that of LC-OCT for identifying different patterns of keratinocyte skin tumors [38]. They concluded that both methods can detect features like erosion and ulceration as well as a disrupted epidermal layer, and LC-OCT had more specificity in recognizing parakeratosis, dyskeratosis, and an irregular vascular arrangement [38].

A study conducted by Soglia et al. used the 3D LC-OCT for the first time to describe the characteristics of MCC in a single case [39]. The features detected through LC-OCT were compared to the ones analyzed through RCM. The LC-OCT features were illustrative of the histological aspect of the tumor. The epidermic layer showed hyperkeratosis and a consistent demarcation from the cutaneous tissue created by the dermal expansion [39]. LC-OCT depicted the affected dermis as white lines and dim areas, reflecting the clusters of Merkel cells enclosed by collagen septa. Within the hyporeflective nests, individual bright cells that were highlighted with RCM were also identified [39]. The study concluded that LC-OCT can prove to be a valuable instrument in the differential diagnosis of MCC as well.

LC-OCT appears promising for early diagnosis of MCC due to its precision on a cellular level, its penetration depth, and three-dimensional perspective. This enables differentiation from other tumors that may present similar clinical patterns [39].

## 6. Conclusions

Considering the aggressive behavior and unfavorable prognosis of MCC, early diagnosis is crucial for improving its outcome. However, limited evidence is available on the clinical and imaging aspect of the tumor at an early stage, as MCC is rare and usually diagnosed late because of its unspecific clinical characteristics.

Since dermoscopy is the main tool used to support clinical diagnosis, further research on MCC dermoscopic patterns is required. Furthermore, advancements in new imaging methods appear promising for differential diagnosis from other tumors, especially benign ones, the assessment of the MCC growth, a better pre-operative estimation of its extension, and the early detection of relapse or cutaneous metastasis. Due to the rarity of MCCs, exploration of the implementation of these techniques in clinical practice is difficult and will require further wide, multicentric cooperation.

## Figures and Tables

**Figure 1 cancers-16-03586-f001:**
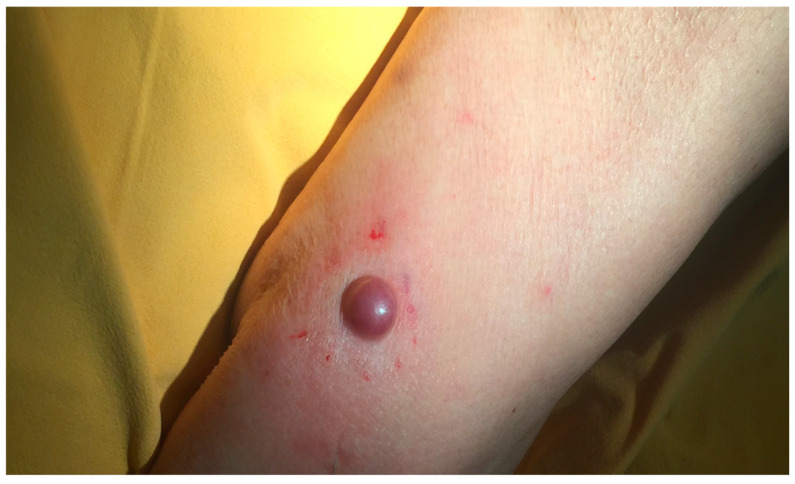
A nodular pink lesion located on the forearm. Courtesy of Prof. Caterina Longo.

**Figure 2 cancers-16-03586-f002:**
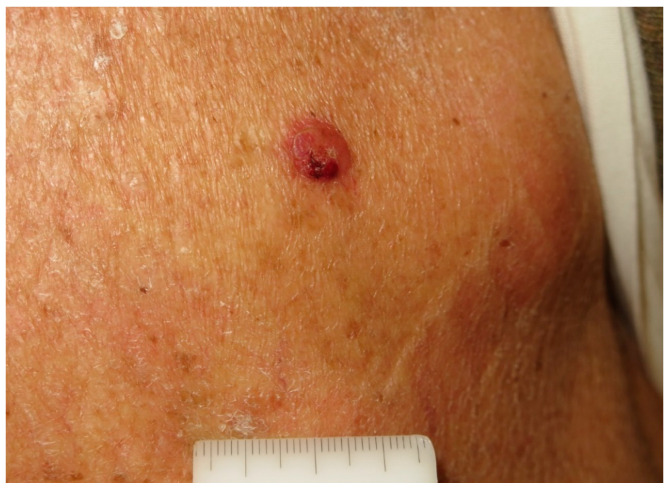
A partially ulcerated pink nodule on the back of a 78-year-old man. Courtesy of Prof. Caterina Longo.

**Figure 3 cancers-16-03586-f003:**
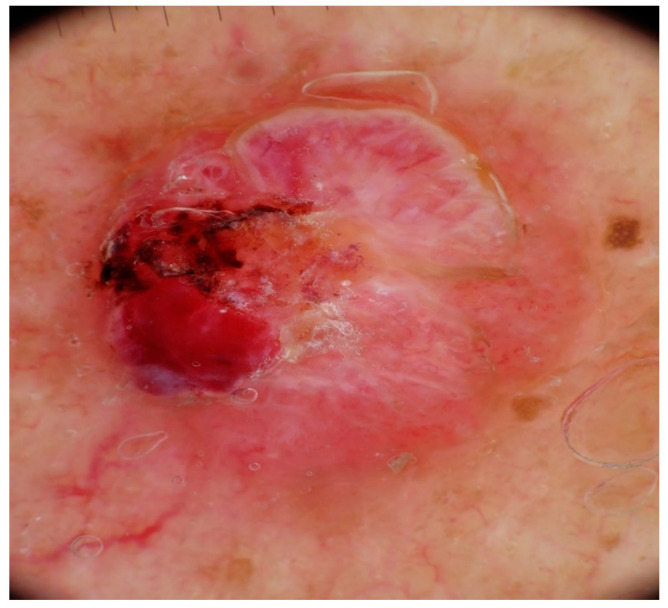
Dermoscopy shows an asymmetric pattern with ulceration, polymorphous vessels, pinkish to cherry-red color, and shiny white structures. Courtesy of Prof. Caterina Longo.

**Figure 4 cancers-16-03586-f004:**
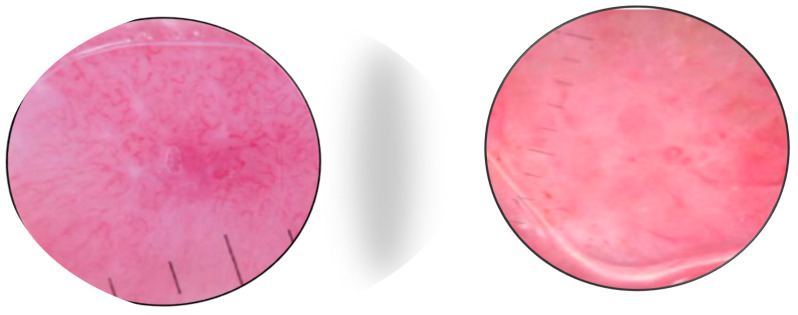
Dermoscopy shows polymorphous vascular patterns, milky red areas on a white background, and arborescent large-caliber vessels. Courtesy of Prof. Ana-Maria Forses.

**Figure 5 cancers-16-03586-f005:**
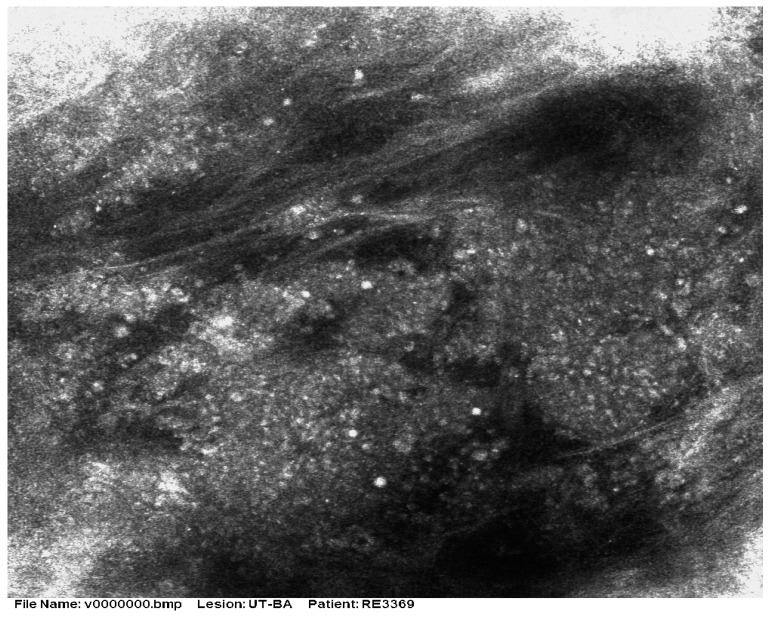
RCM shows the presence of aggregates of small monomorphic cells surrounded by fibrotic septae. Courtesy of Prof. Caterina Longo.

**Figure 6 cancers-16-03586-f006:**
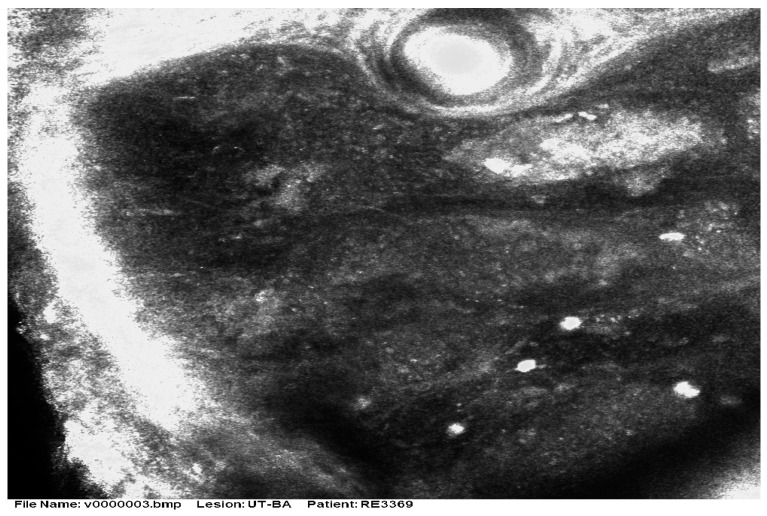
RCM shows small hyporeflective cells arranged to form solid aggregates surrounded by non-reflective stroma. Courtesy of Prof. Caterina Longo.

**Table 1 cancers-16-03586-t001:** Non-invasive imaging techniques used for diagnosis of MCC: advantages and limitations.

Imaging Technique	Advantages and Limitations
**Dermoscopy**	- Higher specificity (89%) than clinical examination (67%) for non-pigmented nodules - Non-invasive visualization of MCC features
**High-frequency ultrasound**	- Cost-effective and non-invasive - Ability to evaluate tumor thickness and guide excision plan - Useful in detecting metastases and assessing lymph nodes
**Reflectance confocal microscopy**	- Limited depth of penetration due to nodular nature of most tumors - Effective for both primary and secondary tumors
**Optical coherence tomography**	- Useful for differential diagnosis with other tumors, including SCC and BCC - Superior in early diagnosis due to cellular level precision and three-dimensional perspective

**Table 2 cancers-16-03586-t002:** Key features of MCC in dermoscopy.

Dermoscopy Features of MCC
- Prominent red hue indicating blood vessels or erythema
- Milky red areas and vascular anomalies (e.g., linear irregular vessels, dotted vessels, polymorphous vascular pattern)
- Pink structureless areas, which can appear as a pink base or smaller rounded areas
- White areas, scales, and hyperkeratosis
- Vascular morphological structures like arborizing or branching vessels (differential diagnosis with BCC)
- Lack of pigmentation or blue-gray veils (differential diagnosis with amelanotic melanoma)

**Table 3 cancers-16-03586-t003:** Key features of MCC in HFUS.

MCC Features in High-Frequency Ultrasound
- Ill-defined dermal lesion with subcutaneous infiltration
- Hypoechoic mass with hyperechoic areas
- Posterior acoustic enlargement
- Dense epidermis
- Highly vascular tumor with Doppler analysis
- Discrete flow in recurrent lesions

**Table 4 cancers-16-03586-t004:** Key features of MCC in RCM.

MCC Features in Reflectance Confocal Microscopy
- Grouped hyporeflective cells with few bright cells in fibrotic stroma
- Visible blood vessels next to tumor proliferation
- Dermic clusters of compact hyporeflective cells with connective fibrous tissue bands
- Disrupted epidermis with low density
- Monomorphous appearance with defined fibrotic stroma (differential diagnosis with amelanotic melanoma)
- Dissociation of cell aggregates in certain tumor regions (corresponding to histology)

**Table 5 cancers-16-03586-t005:** Key features of MCC in OCT and LC-OCT.

MCC Features in Optical Coherence Tomography
- Hyperkeratosis and clear demarcation from cutaneous tissue
- Affected dermis shown as white lines and dim areas (clusters of Merkel cells in collagen septa)
- Hyporeflective nests with individual bright cells
- High-resolution cross-sectional images of skin layers with 2 mm depth
- High-definition OCT (HD-OCT) enables 3D visualization and differentiation of BCC subtypes
- Line-field confocal coherence tomography (LC-OCT) shows features like hyperkeratosis, epidermal disruption, and vascular pattern

## Data Availability

No new data were generated or analyzed in this study. Data used in this review are available from the original published sources cited in the article. The authors confirm that this material is original and has not been published in whole or part elsewhere and is not currently being considered for publication in another journal.

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
