# Peer review of "A Review of Non-Invasive Skin Imaging in Merkel Cell Carcinoma: Diagnostic Utility and Clinical Implications"

_cancers, 2024, doi:10.3390/cancers16213586_

Round 1

Reviewer 1 Report

Comments and Suggestions for Authors

The present paper is a narrative review about non-invasive skin imaging methods that can serve as additional tools in the examination of Mekel cell carcinoma (MCC), which is a are and aggressive cutaneous neuroendocrine malignancy characterised by its propensity for rapid growth and early regional and distant metastasis. The paper is well written in English and it presents an informative state of the art, which is pertinent to introduce the problem.

My main concerns with the present research are the following:
a) Please reduce the size of Figures 1 and 2, in order to add more MCC samples.
b) For each section (2 Dermoscopy,3 HFUS,4 RCM,5 OCT):
i) Please for each section, share a link or website where the scientific community. can explore a database of dermoscopy images.
ii) Please present information about advantages and disadvantages.
iii) Please present information about the commonly used software for image analysis.
iv) Please present relevant information about how the medical device generates the corresponding image..
c) Please present relevant information about which imaging method must be used according to different patterns of MCC.

Comments on the Quality of English Language

The paper is well written in English.

Author Response

Reviewer #1: The present paper is a narrative review about non-invasive skin imaging methods that can serve as additional tools in the examination of Mekel cell carcinoma (MCC), which is a are and aggressive cutaneous neuroendocrine malignancy characterised by its propensity for rapid growth and early regional and distant metastasis. The paper is well written in English and it presents an informative state of the art, which is pertinent to introduce the problem.

Reply: Thank you for your kind words!

My main concerns with the present research are the following:
a) Please reduce the size of Figures 1 and 2, in order to add more MCC samples.

Reply: Thank you for the suggestion, we have modified the size of these figures

  1. b) For each section (2 Dermoscopy,3 HFUS,4 RCM,5 OCT):
    i) Please for each section, share a link or website where the scientific community. can explore a database of dermoscopy images.

Reply: I understand and appreciate your suggestion, however there are very few cases of Merkel Cell Carcinoma, given the fact that it is a very rare tumor, therefore there is no available dermoscopy database yet. However, a part of the authors are working on an ongoing study coordinated by IDS (International Dermoscopy Society) where they are collecting cases of Merkel Cell Carcinoma dermoscopy. The tumor is very rare and the biggest case series published in dermoscopy consists of 12 cases.

  1. ii) Please present information about advantages and disadvantages.

Reply: One of the tables included in the article presents advantages and limitations of each method. (Table 1) We have included the tables and figures in the final manuscript.

iii) Please present information about the commonly used software for image analysis.
iv) Please present relevant information about how the medical device generates the corresponding image..

Reply: Thank you for your suggestion! We have added further information about the functioning of each non-invasive imaging method.
c) Please present relevant information about which imaging method must be used according to different patterns of MCC.

Reply: Given the rarity of Merkel Cell Carcinoma, the purpose of these methods is for the clinician to recognize the lesion as malignant, therefore becoming an additional tool for clinical assessment and further decision to biopsy. All things given, it is impossible to provide such information because there are no specific features in any of the methods.

Reviewer 2 Report

Comments and Suggestions for Authors

I have read the manuscript carefully, and while the topic is important, there are some key points that should be addressed by the authors before publication.

The introduction is fine and sets out very well the context and the authors' aim. But the manuscript has no section explaining the methodology used, there is no indication of criteria for the selection of references or sources of information, which is important for a review article. There should be a section specifying the databases used, the keywords used, the inclusion and exclusion criteria for the references and the time frame for the literature search.

In the abstract, Raman spectroscopy and fluorescence polarisation techniques are mentioned that can be used for this purpose. However, these methods are not mentioned in the body of the article, which may confuse readers. I suggest removing them from the summary or explaining them in detail in the text.

It would be good if the basic principles of each imaging technique were explained in each section, even if only briefly. This is especially important for readers who are not familiar with the individual techniques, and a brief explanation of how each method works would be very instructive for the manuscript.

The authors have provided several images, but do not identify the source for any of them. All images used should be properly cited. This includes the source of the image and permission to use it.

Overall, the manuscript is worthy of publication, but some aspects still need to be corrected before it can be considered for publication.

Author Response

Reviewer #2: I have read the manuscript carefully, and while the topic is important, there are some key points that should be addressed by the authors before publication.

The introduction is fine and sets out very well the context and the authors' aim. But the manuscript has no section explaining the methodology used, there is no indication of criteria for the selection of references or sources of information, which is important for a review article. There should be a section specifying the databases used, the keywords used, the inclusion and exclusion criteria for the references and the time frame for the literature search.

In the abstract, Raman spectroscopy and fluorescence polarisation techniques are mentioned that can be used for this purpose. However, these methods are not mentioned in the body of the article, which may confuse readers. I suggest removing them from the summary or explaining them in detail in the text.

It would be good if the basic principles of each imaging technique were explained in each section, even if only briefly. This is especially important for readers who are not familiar with the individual techniques, and a brief explanation of how each method works would be very instructive for the manuscript.

The authors have provided several images, but do not identify the source for any of them. All images used should be properly cited. This includes the source of the image and permission to use it.

Overall, the manuscript is worthy of publication, but some aspects still need to be corrected before it can be considered for publication.

Reply: Thank you for appreciating our manuscript. Following the suggestion, we have removed the two techniques mentioned in the abstract. Moreover, we have added further explanations on the basic principles of the techniques in each section.

Finally, the images belong to the authors (Courtesy of Ana-Maria Forsea and Caterina Longo), they are original and have not been used before.

Our work is based on a search of the literature performed in PubMed for scientific evidence, including all studies regarding Merkel Cell Carcinoma and the non-invasive methods described below that were used for diagnostic purposes between 2000 and 2024. Keywords selected were: “Merkel cell Carcinoma” and any of the following: “diagnosis, imaging, high frequency ultrasound, dermoscopy, confocal microscopy, ultrasound, optical coherence tomography, multispectral, non-invasive, imaging”. The search was limited to works in English, and to studies in human subjects. Titles and abstracts were checked for eligibility. References of selected relevant articles were retrieved and included in analysis if relevant.

Round 2

Reviewer 1 Report

Comments and Suggestions for Authors

I understand the problem to provide specific features because of the rarity of Merkel Cell Carcinoma.

My comments have been properly addressed.

Comments on the Quality of English Language

The paper is well written in English.

Reviewer 2 Report

Comments and Suggestions for Authors

After careful reading of the revised manuscript and the author's response to the reviewer's comments, the authors have addressed the reviewer's concerns and have made appropriate changes to improve the quality of the manuscript. I have no further comments. I only recommend a final thorough text check as some typos have crept in, e.g. on page 6 where it says “20 MHz -50 MH” it should read “20 MHz -50 MHz”. Recommendation for publication.